# Multiclass Level-Set Segmentation of Rust and Coating Damages in Images of Metal Structures

**DOI:** 10.3390/s22197600

**Published:** 2022-10-07

**Authors:** Michał Bembenek, Teodor Mandziy, Iryna Ivasenko, Olena Berehulyak, Roman Vorobel, Zvenomyra Slobodyan, Liubomyr Ropyak

**Affiliations:** 1Department of Manufacturing Systems, Faculty of Mechanical Engineering and Robotics, AGH University of Science and Technology, 30-059 Kraków, Poland; 2Department of the Theory of Wave Processes and Optical Systems of Diagnostics, Karpenko Physico-Mechanical Institute of the NAS of Ukraine, 5 Naukova St., 79060 Lviv, Ukraine; 3Department of Computer Sciences, University of Lodz, Pomorska Str. 149/153, 90-236 Lodz, Poland; 4Department of Corrosion and Corrosion Protection, Karpenko Physico-Mechanical Institute of the NAS of Ukraine, 5 Naukova St., 79060 Lviv, Ukraine; 5Department of Computerized Engineering, Ivano-Frankivsk National Technical University of Oil and Gas, 76019 Ivano-Frankivsk, Ukraine

**Keywords:** level-set method, color image processing, coating damage, rust detection, multiclass image segmentation

## Abstract

This paper describes the combined detection of coating and rust damages on painted metal structures through the multiclass image segmentation technique. Our prior works were focused solely on the localization of rust damages and rust segmentation under different ambient conditions (different lighting conditions, presence of shadows, low background/object color contrast). This paper method proposes three types of damages: coating crack, coating flaking, and rust damage. Background, paint flaking, and rust damage are objects that can be separated in RGB color-space alone. For their preliminary classification SVM is used. As for paint cracks, color features are insufficient for separating it from other defect types as they overlap with the other three classes in RGB color space. For preliminary paint crack segmentation we use the valley detection approach, which analyses the shape of defects. A multiclass level-set approach with a developed penalty term is used as a framework for the advanced final damage segmentation stage. Model training and accuracy assessment are fulfilled on the created dataset, which contains input images of corresponding defects with respective ground truth data provided by the expert. A quantitative analysis of the accuracy of the proposed approach is provided. The efficiency of the approach is demonstrated on authentic images of coated surfaces.

## 1. Introduction

The use of protective coating is the most common way to prevent damage to the surface of different construction materials. The type of protective coating depends on the material itself and the environment it operates in. Many studies were conducted to investigate their properties and ways of monitoring their state during operation. The availability of metal, oxide, composite and non-metallic coatings allows for rationally combining the base and coating materials’ properties. The most frequently used coatings are paint coatings [1,2,3,4,5], polymers [6,7,8], and biopolymer films [9,10]. All these types of widespread coatings are used in various fields of technology and medicine. In addition, paint coatings are also used as sensors to determine the temperature and coefficient of thermal conductivity [11,12], pressure, and deformations in the studied samples [13,14].

Design, technological, and operational methods are used to provide long-term operation of coated products. Design methods include the selection of base and coating materials, conduction of corrosion [15,16,17], and tribocorrosion tests [18]. In particular, the scientists [15,16] apply microelectrochemical studies of the stressed state of the metal on its structural components and crack-like defects, and in [17], the corrosion behavior of coated carbon steels in salt water is investigated. Porosity studies of coatings [19] and tribological tests are carried out using computer image processing [20,21,22]. For example, researchers [20] used the image processing technique to develop a method of tribotechnical testing of coatings under conditions of wear by unfixed abrasive; and in works [21,22], a photogrammetric approach was used to study the wear process and estimate volume loss. Strength tests [23,24] as well as modeling and performance of thermal calculations of coatings [25,26], are also performed.

Analytical methods of stress analysis in layered coatings under local load [27,28,29], studies of the influence of flexible [30,31] and composite [32,33,34] coatings on the limited state of cracked plates and shells are worth considering. In addition, the coating of damaged surfaces can be treated as the healing of cracks by injection technologies with pliable aggregates [35,36,37] or non-contrast materials [38,39].

It is necessary to consider technological heredity to ensure the product’s functioning throughout its life cycle when preparing the surface for coating [40,41,42]. Technologies for applying environmentally friendly hydrophobic, photocatalytic, and antimicrobial coatings that do not cause environmental pollution are being developed [43,44,45]. Operational methods include surface cleaning and repair of damaged areas of coatings. It is crucial to establish a reasonable period for carrying out such works.

Automating different types of object surface inspection tasks draws more attention to developing image processing techniques. Such automation usually implies the acquisition of inspected object images with their subsequent computer processing. Object images can be taken remotely, avoiding direct visual inspection. The growing accessibility of autonomous drones allows the visual inspection of distant, large, or hard-to-reach objects. One such inspection task is the monitoring of the state of coated surfaces. In this scope, we mainly focus on painted steel surfaces of infrastructure objects (oil tanks, bridges, high voltage power lines towers, etc.). Such coated surfaces can be subjected to various possible damages that may lead to further structural degradation and, if detected, ought to be dealt with. In this work, we restrict our research to three types of such damages. Paint cracks and flaking are two coating-related damages, and the third is rust damage affecting steel surfaces.

Examples considered in this paper damages are shown in Figure 1. Paint cracking is the defect of damage to the structural integrity of paint coating in the form of a crack that allows access to air and humidity on the steel surface. Flaking is the damage where some local region of the steel surface loses its protective paint coating but has no signs of rust damage. Moreover, finally, rust damage is a defect where the local region of the steel surface loses its protective paint coating and is affected by corrosion.

Metal rusting usually causes the protective coating to break down and thus leaving distinguishable marks on the inspected object’s surface. The significant contrast between rust damage and protective coating colors allows the segmentation of the damaged areas based on the color features only. Numbers of articles were dedicated to the segmentation of color images. Many of them were using *k*-means [46], Gaussian mixture based models (GMM) [47], graph-based approaches [48,49], neural networks [50,51,52,53], support vector machines (SVM) [54,55,56], shape index [57], level set models [58,59] etc. A significant amount of research has been devoted to rust images.

Digital image processing and pattern recognition are powerful tools in non-destructive testing. A comprehensive survey of the application of image processing to assess material degradation was made by Xia et al. [60]. The authors started with a description of basic image types and standard image pre-processing methods. Considerable attention was focused on the processing methods of corrosion images, in particular model-based measurements, structural methods, statistical parameters, co-occurrence of matrix-based features, and transform-based models. The assessment of material degradation, including different types of corrosion (pitting, uniform, undercoating, stress cracking) and degradation of the organic coatings, was considered. The cooperation of image processing with other rust detection methods was reviewed. Modern challenges in rust detection using image processing are listed, and the ways of problem solutions, especially by artificial intelligence, machine learning, and deep learning, are outlined.

Jardim et al. [61] proposed an approach to color image segmentation consisting of region extraction by successive combining of *k*-means clustering and watershed methods. The experimental results and comparison with other methods are provided on both graphic and natural images. The proposed algorithm is stated to cope well with image databases with high variability.

Cluster-weight and group-local feature-weight learning in fuzzy C-means algorithm (CGFFCM) was proposed to cope with image segmentation task [62]. This method is based on clustering. The features from three groups (local homogeneity, color, and structure) are extracted and used by CGFFCM. The imperialist competitive algorithm (ICA) optimizes the weighted feature process. The method applies to images of different semantics.

Katsamenis et al. developed a method for corrosion localization and classifying rust grades based on a U-Net convolutional neural network [63]; the proposed method consists of three steps. The first step is the segmentation of corroded regions; the second is contour refining using the data projection scheme, and the third is the rust grade classification. Experienced engineers segmented and assigned corrosion grades to 400 images. They were used for training the U-Net models of the first layer. The performance was evaluated using such metrics as accuracy, precision, recall, F1-score, and intersection over the union. The proposed SLPAC U-Net demonstrated improvement in comparison with *k*-means and GMM methods.

Khayatazad et al. detected corrosion regions on the steel structures using roughness and color [64]. They measured image uniformity on the base of the grey-level co-occurrence matrix. Color images were converted to greyscale with 8 levels of grey. The authors used the HSV color space and normalized histogram of H and S to describe corrosion colors. The performance was evaluated using recall and precision metrics. The training dataset contained eight images. Authors categorized corroded areas in images with an average recall of 85%.

Not many studies have analyzed defects different from rust on painted steel surfaces using image processing. For example, Zhang et al. used Michelson interferometer for paint flakes detection and modification of Chan and Vese’s method of flake segmentation [65].

Rezaie et al. demonstrated the advantages of a deep convolutional neural network over thresholding the maximum principal strain map for segmenting cracks in the images of stone walls [66].

Abas and Martinez used mathematical morphology and grid-based automatic thresholding for crack segmentation. Classification is based on such features as orientation and length of line segments [67].

A crack detection approach to monitoring the civil infrastructure is proposed [68]. Technical elements are discussed to optimize the image acquisition process. The morphological operator was used for crack-like pattern segmentation. The feature extraction process was fulfilled by linear discriminant analysis (LDA). The neural networks, SVM, and a nearest-neighbor classifier were used. The representation of detected cracks was done with a multi-scale crack map. The proposed approach was tested on the database of 220 real concrete cracks and 200 non-crack images.

A novel deep crack segmentation network was proposed [69] to succeed in two mutually exclusive tasks: increasing speed and accuracy. The feature extraction consists of two directions—morphological and shallow detail. The proposed method’s effectiveness in comparison with state-of-the-art CNN-based networks was justified based on three public datasets.

Our previous works [70,71,72,73,74] were dedicated to rust damage segmentation under conditions that can distort its damage percentage assessment. They were focused on the detection of one type of damage—rust. In particular, rust detection based on HSV image color model [71], the use of the single-scaled retinex method [72], the influence of color restoration based on a color checker [73], and the reduction of shadow effects [74] algebraic model with an asymmetric characteristic [75] and methods of logarithmic type image processing [76] as well as the application of inhomogeneity inforced piecewise smooth model [77] for image segmentation were considered. A dimensionality reduction PCA technique was applied in [78] for the rust segmentation in images obtained in irregular lighting conditions.

The majority of reviewed papers are dedicated to the detection of certain types of defects. Most of them focus purely on rust detection or paint damage separately. In the case of real-world applications of such approaches to regular inspection of industrial structures, it is reasonable to detect simultaneously protective coating damages. The reason for that is that even in case of the absence of explicitly visible rust damage, there can be damage to the protective paint coating. Such coating damages are the signs of possible future rusting of the steel surface. Thus detection and treatment of such defects in the early stages can prevent further rusting of the underlying steel surface.

The most common defects of protective coating (relevant for developing rust damage) are paint cracks and paint flaking. Thus, this work aims to develop the approach for detecting multiple damages (of the steel surface and protective coating) within one computational framework. To achieve this goal, the following tasks should be solved: construction of the optimization multiclass segmentation method based on a new regularizer; application of advanced techniques on the image dataset; comparison of segmentation results with ground truth data and calculation of segmentation accuracy.

The main innovation of this paper is the introduction of the label-dependent penalty term that incorporates segmentation results of multiclass SVM and valley detection methods for refined, resulting multiclass segmentation. To the best of our knowledge, it is the first time that three types of damage (coating crack, coating flaking, and rust damage) are detected within one computational framework.

## 2. Materials and Methods

The proposed approach suggests a combination of machine learning and image processing methods for multiclass segmentation of different defect types. Development, testing and validation of the considered approach were done based on the dataset of images of different defect types.

We used a set of 150 images with a resolution of 480 × 640. The images in the database show coatings of different colors, as well as different types of their damage—coating crack, coating flaking, and rust damage. There are images with one type of defect presented on it and images with multiple types of damage in the dataset. Image examples are presented in Figure 1. Painted steel surfaces (low carbon structural steel St3 State Standard GOST 380–2005 with glyphtal enamel as a protective coating) of infrastructure objects of long-term operation in a moderate climatic zone in the open air were analyzed. All images were acquired in conditions of natural lightning. Images were saved in the RGB color space. Digital camera Canon PowerShot S95, ISO-speed ISO80, 10.0 megapixels was used with sRGB color gamut and 6.0–22.5 mm focal length.

A described set of images was used to create a dataset to train the proposed model and validate obtained results. The training dataset consists of several acquired input images with respective ground truth data. Ground truth data consist of manually segmented input images under the guidance of an expert with more than 40 years of experience in corrosion engineering. Then we made the quantitative comparison of the developed method with an expert’s evaluation. We calculate an error in the segmentation map obtained by the developed method compared to ground truth data.

The proposed approach is shown schematically in Figure 2.

The proposed method consists of two stages that are briefly summarized below.

The first stage is an initial segmentation of the input image. For this purpose, two different approaches are used:(a)The SVM is trained in a supervised learning mode based on a previously labeled training set with ground truth data for rust, flacking, and background classification. SVM was chosen for its ability to build an optimal decision hyperplane that separates classes. In this work, we use a multiclass SVM with a one against all approach. It is used as a simple segmentation method only for three-class classification (background, rust, and flaking). These three classes are discriminated by multiclass SVM because their features do not overlap in RGB color space.(b)We used a different approach for the primal segmentation of paint cracks. The reason for that is that they do not have unique color features. As a result, paint cracks in RGB color space overlap with the other three classes making color features insufficient for classification. The characteristics that distinguish paint cracks are more of a geometric nature. We use a modified “valleys” detection method as a primal segmentation method for that purpose.

The second stage of the approach suggests the fusion of outputs of the first stage to produce the final segmentation map.

In this work level-set approach is used to aggregate results of the initial segmentation of background, rust, and flaking by multiclass SVM and segmentation of paint cracks by the valley detection method. The proposed in this paper level-set approach uses a specially developed label penalty term to aggregate the results of two primal classifiers and to produce a final segmentation map.

## 3. Results

### 3.1. Level-Set Method for Image Segmentation

The level-set method [79,80] is an optimization-based approach widely used for image processing tasks. It is used in variety of different formulations [80,81,82,83,84,85,86] and solves number of image processing problems [87,88,89]. It allows the formulation of image processing tasks in terms of energy minimization. Therefore it is possible to design different constraints and properties imposed on a developed image processing method.

Let $u_{0}:\Omega \to ℜ$ it be a greyscale image and $\Omega \in {ℜ}^{2}$ be an image domain. The level-set method allows representing the image partition using a level-set function (LSF) $\varphi \left(x,y\right)$ where $\left(x,y\right)\in \Omega$ spatial coordinates are used. The contour C that divides the input image $u_{0}$ into disjoint segments is represented via LSF $\varphi \left(x,y\right)$. More formally C is represented by a zero level-set of $\varphi \left(x,y\right)$ and defined as $C=\left\{\left(x,y\right)\in \Omega :\;\varphi \left(x,y\right)=0\right\}$. Thus, partitioning the overall image domain Ω into two disjoint regions is defined as follows $\Omega_{1}=\left\{\left(x,y\right):\;\varphi \left(x,y\right)>0\right\}$$\Omega_{2}=\left\{\left(x,y\right):\;\varphi \left(x,y\right)<0\right\}$.

Modern level-set developments take their origin from the research by Mumford and Shah [79]. In general, the solution of the Mumford–Shah problem is a complicated task that makes it less flexible for various image processing tasks.

Chan and Vese [80] simplified the original problem [79] by introducing a variational formulation of the original Mumford-Shah functional for a limited number of image partitions. For the two-class image segmentation Chan and Vese’s functional looks as follows
(1)$$\begin{array}{l} F\left(c_{1},c_{2},\varphi \right)=\mu \int_{\Omega }\left|\nabla H\left(\varphi \right)\right|dxdy+\lambda \int_{\Omega }{\left|u_{0}\left(x,y\right)-c_{1}\right|}^{2}H\left(\varphi \right)dxdy+\lambda \int_{\Omega }{\left|u_{0}\left(x,y\right)-c_{2}\right|}^{2}\left(1-H\left(\varphi \right)\right)dxdy\;, \end{array}$$
where μ and λ are coefficients,$H\left(\varphi \right)$ represent Heaviside function, $c_{1}$ and $c_{2}$ represent the mean of greyscale values for $\Omega_{1}$ and $\Omega_{2}$, respectively. In [90] authors developed an additional penalty term $R_{p}=\nu \int_{\Omega }p\left(\left|\nabla H\left(\varphi \right)\right|\right)dxdy,$ where ν is the coefficient, p is the potential function defined as $p\left(s\right)=\frac{1}{2}{\left(s-1\right)}^{2}.$ It forces LSF φ(x,y) to maintain properties of the signed distance function for better convergence. With additional penalty term $R_{p}$ expression (1) can be represented in terms of energy minimization as follows
(2)$$E=E_{data}+E_{c}+R_{p},$$
where $E_{data}=\lambda \sum_{i=1}^{N}\int_{\Omega }e_{i}M_{i}\left(x,y\right)dxdy$, $E_{c}=\mu \int_{\Omega }\left|\nabla H\left(\varphi \right)\right|dxdy$, N is a number of segmentation classes, $e_{i}={\left|u_{0}\left(x,y\right)-c_{i}\right|}^{2}$ and $M_{i}\left(x,y\right)$ is a membership function $M_{i}(x,y)=\left\{\begin{array}{c} 1,\;\mathrm{if}\;(x,y)\in \Omega_{i}\\ 0,\;\mathrm{if}\;(x,y)\notin \Omega_{i} \end{array}\right.$, which, in the case of (1), is defined through $H\left(\varphi \right)$ as $M_{1}\left(x,y\right)=H\left(\varphi \right)$, $M_{2}\left(x,y\right)=1-H\left(\varphi \right)$.

The term $E_{c}$ acts as a penalty for the length of the contour C, to ensure its smoothness. While $E_{data}$ is a data term that acts as a similarity measure of an image $u_{0}$ and its model is represented by model parameters $c_{i}$. Thus segmentation of the image $u_{0}$ is accomplished by finding such function φ(x,y) and constants $c_{1}$ and $c_{2}$ that minimize the energy functional E (2).

The common approach to energy functional E minimization is through the solution of the Euler–Lagrange equation that leads to the following
(3)$$\frac{\partial \varphi }{\partial t}=-\frac{\partial E}{\partial \varphi }.$$

Thus the solution to the minimization problem of (1) in terms of (3) is as follows
(4)$$\frac{\partial \varphi }{\partial t}=\nu \left[\nabla^{2}\varphi -div\left(\frac{\nabla \varphi }{\left|\nabla \varphi \right|}\right)\right]+\mu \delta_{\varepsilon }(\varphi )div\left(\frac{\nabla \varphi }{\left|\nabla \varphi \right|}\right)+\delta_{\varepsilon }(\varphi )\left(e_{1}-e_{2}\right),$$
where $\delta_{\varepsilon }\left(z\right)=\frac{1}{\pi }\frac{\varepsilon }{\varepsilon^{2}+z^{2}}$ is the regularised representation of Dirac delta function In the case where the image segmentation task requires the use of more complicated features than greyscale pixel values, it was proposed in [91] to use the negative logarithm of the probability density function as a data term
(5)$$e_{i}=-\log\left(p_{i}\left(x,y\right)\right),$$
where $p_{i}\left(x,y\right)$ represents the probability density function that describes the model of i-th class.

In this work, we consider the image segmentation task for the case of four classes (N=4). The distinguished classes present three types of damage described above with undamaged coating being the fourth one. Originally in [81], segmentation of multiple classes by (1) requires the use of more than one LSF and is referred to as the multiphase level-set approach. Extension of the model (1) for four class cases requires the usage of two LSFs.

Instead of a classical multiphase level set, we used a slightly different approach for our work. Similarly to [92], we used multiple level sets—a separate LSF for every class. Their codependent optimization is performed through the so-called label term Ψ.

The described model is flexible and allows the construction of multiclass segmentation models with various data- and penalty terms. Such properties make it suitable for the design of effective image segmentation techniques.

### 3.2. Energy Terms for Defect Segmentation

The color of rust is mainly defined by the color properties of its chemical components and occupies a certain domain in the color space. It makes it possible to build a segmentation method using RGB features only for the rust segmentation task. Background and such defects as paint flaking also can be separated by their color features only. For that purpose we used three-dimensional feature vector (r,g,b), where r, g and b are “red”, “green”, and “blue” values of pixel color components from the RGB color model, respectively.

Paint cracking has to be segmented by a different method. The main reason for that is that the color features of paint cracking can be similar to rust damage or steel surface. Thus cracking requires additional features to be used for successful segmentation. Those features should account structural characteristics of the cracks. For this purpose, we used a well-known valley detection approach based on scale-space representation [93].

Unfortunately, there is an overlap (in feature space) between regions with flaking, rust damage and paint cracking. The main reason is that the cracking region color can be close to either rust color or color of the steel surface. Such overlapping leads to the additional error of segmentation due to emerging ambiguity.

One of the main contributions of this article is the introduction of the label-dependent penalty term Ψ. It allows incorporation segmentation results for two different classifiers (multiclass SVM and valley detection method) and resolves the problem of their overlapped segmentations. The proposed penalty term Ψ is defined as follows
(6)$$\Psi =\sum_{i=1}^{N}\sum_{j=1|\left(i\neq j\right)}^{N}\Psi_{ij}\;,$$
where $\Psi_{ij}=\int_{\Omega }M_{i}M_{j}G_{ij}dxdy$; $M_{i}$ and $M_{j}$ are indicator functions and are defined as $M_{i}=H\left(\varphi_{i}\right)$ and $M_{j}=H\left(\varphi_{j}\right)$; $G_{ij}$ is an intersection potential to be defined below. The derivative of $\Psi_{ij}$ is defined as
(7)$$\frac{\partial \Psi_{ij}}{\partial \varphi_{i}}=\delta_{\varepsilon }(\varphi_{i})\;G_{ij}H\left(\varphi_{j}\right).$$

After initialization of all LSF $\varphi_{i}$ based on the results of preliminary segmentation methods (SVM and valley detection), there is no need for further use of data term $E_{data}$ in (2). Given all the above we introduce the final model as follows
(8)$$E=E_{c}+R_{p}+\Psi .$$

One should observe that solution (4) of minimization problem (1) in terms of (3) contains the gradient of the function. The solution of (8) contains the gradient as well. It characterizes the function’s behavior along the direction in which it is increasing most rapidly. But it is possible to replace the gradient with a directional derivative starting from (1) and to study a similar problem along a direction. There are many papers analyzing the influence of directional derivatives on the properties of such functions [94,95,96].

The solution to (8) is the next
(9)$$\frac{\partial \varphi_{i}}{\partial t}=\nu \left[\nabla^{2}\varphi_{i}-div\left(\frac{\nabla \varphi_{i}}{\left|\nabla \varphi_{i}\right|}\right)\right]+\mu \delta_{\varepsilon }(\varphi_{i})div\left(\frac{\nabla \varphi_{i}}{\left|\nabla \varphi_{i}\right|}\right)+\sum_{j=1|\left(i\neq j\right)}^{N}\delta_{\varepsilon }(\varphi_{i})\;G_{ij}H\left(\varphi_{j}\right)\;.$$

The product $M_{i}M_{j}$ in (6) defines the intersection of i-th and j-th segments. We define the intersection potential $G_{ij}$ as follows
(10)$$G_{ij}=\sum_{k=1}^{N}\sum_{l=1|\left(k\neq l\right)}^{N}P_{kl}\left(M_{i},M_{j}\right)\;,$$
where the function $P_{kl}$ is defined for every intersection $M_{i}M_{j}$ separately. In our case, we have to consider only three intersection cases: the intersection of crack segmentation with the segmentation of all other classes. This is because we can have ambiguous segmentation only of paint crack (which is segmented with a valley detection approach) and other classes: rust, flacking, and background (which are defined by multiclass SVM and thus do not intersect). Therefore non-zero are only $\Psi_{14}$, $\Psi_{41}$, $\Psi_{24}$, $\Psi_{42}$, $\Psi_{34}$ and $\Psi_{43}$. Here values of indexes i and j represent respective classes as follows: 1—“rust”, 2—“paint”, 3—“flaking”, 4—“paint crack”. As mentioned previously, the crack defect poses the main problem in terms of segmentation error based strictly on color features. Thus the main attention was paid to the design of $\Psi_{ij}$ for this type of defect, which in turn is fully defined through values of $P_{kl}$. For our segmentation goals, we designed the necessary $P_{kl}$ in the following way
(11)$$\left\{\begin{array}{l} P_{14}=-s\left(C_{1}-C_{2}\right)+2\left(C_{4}-C_{1}\right),\\ P_{41}=s\left(C_{1}-C_{2}\right)-2\left(C_{4}-C_{1}\right),\\ P_{24}=0.33,\\ P_{34}=s\left(C_{4}-C_{3}\right),\\ P_{43}=-s\left(C_{4}-C_{3}\right), \end{array}\right.$$
where $C_{i}$ is defined as $C_{i}=K*M_{i}$, s is a coefficient, K is an averaging kernel, and ∗ defines convolution operator.

Let us consider the reasoning behind the choice of form of some $P_{kl}$. Every $P_{kl}$ is multiplied by $M_{i}M_{j}$. Therefore it contributes to the model’s evolution only in the area of the intersection of i-th and j-th segments. It is important to construct $P_{kl}$ in a way it would penalise undesirable segmentation as the goal of minimization of (8) is to minimize intersections between any segments $M_{i}$ and $M_{j}$ for i≠j. For example, consider the shape of $P_{34}$ (the intersection of paint flaking and paint crack segments). It depends on values of $C_{i}$, which is average area of class i around every point $\left(x,y\right)$. Values of $C_{3}$ inside flaking area are mostly higher than values of $C_{4}$ thus penalizing the $M_{i}$ in this area. The same logic is true vice versa for values of $P_{43}$. Similar reasoning is applied to construction of all other $P_{kl}$. The only exception is $P_{24}$ which is assigned a constant value and $P_{42}$ is equal to zero. Such values for $P_{24}$ and $P_{42}$ basically mean that in case of intersection $M_{2}M_{4}$ of paint segment and paint crack segment the proposed level-set approach would force this intersection to be segmented as paint. This is because preliminary segmentation of paint is more reliable than segmentation of paint crack (it is because of thresholding in the valley detection approach tends to oversegment the paint crack area).

The penalty term $\Psi_{ij}$ imposes restrictions on some geometrical features of i-th class segment with its neighboring class segments. Those restrictions’ shape depends on the kernel K’s shape and values $P_{kl}$. This term depends on current segmentation labels thus, it is computed at each iteration.

### 3.3. Evaluation of the Performance of the Developed Method of Defect Segmentation

Performance evaluation of the proposed technique is made quantitatively as well as qualitatively. Qualitative assessment is given in the discussion part of the paper. A numerical evaluation of the proposed approach performance is described below and conducted based on validation data set, samples excluded from the training dataset. 

To evaluate the performance of the developed defect segmentation technique we use the receiver operating characteristics (ROC) curves for all classes with the methodology of using one class versus the rest (Figure 3). The ROC curve reflects the statistics of true positive/false positive rates for the binary classifier. In our case, the ROC curve combines four plots for each of the considered here object classes.

The accuracy of the developed approach was estimated in terms of segmentation error rates evaluated on ground truth data. Table 1 contains individual error rates for all classes and the developed approach’s overall segmentation error. Error rates are computed as the percentage of falsely segmented image pixels compared to ground truth data. The table also contains AUC (area under the curve) values computed separately for each class.

Table 2 contains examples of input images (150 × 150 pixels) from the validation set (column 1), the ground truth segmentation validated by an expert (column 2), and the results of segmentation by the developed technique (column 3). This paper’s segmentation maps (ground truth or numerical experiments) are color coded in the following way: red for the area with rust damage, green for background paint, yellow for flaking, and blue for paint cracks.

Examples of segmentation results of authentic images of damage to painted steel surfaces are presented in Table 3. Developed in this paper method for combined rust and coating damage segmentation was tested on a set of authentic images that contained considered defects alone and their combinations.

Results in Table 3 were produced by model (9) with the following parameter values: μ=0.02, ν=0.065, kernel K is the uniform disk shaped averaging kernel with radius r=11 pixels. Table 3 contains input images of differently coated steel surfaces with visible damage of different types. The last column shows the color-coded segmentation results of multiple types of defects.

Table 4 presents examples of segmentation by valley detection technique used in this work. It shows the input image, its valley detection filter response (described in [94]), and the results of applying different global threshold values for valley segmentation. As one can see, by changing the threshold value (as well as filter parameters), we can alter the method’s sensitivity. Thus we can choose how small or vague crack we want to detect. As we can conclude, the chosen type of valley detection method is quite sensitive and can be effectively used in this work. However, it should be noted that the developed approach can incorporate any other crack detection method suitable for the task.

Table 5 illustrates the segmentation results of the input image at different stages of the algorithm functionality. As mentioned earlier, preliminary coarse segmentation of the input image is performed in the first stage. In our case, the input image is segmented by the valley detection method (second image) and multiclass SVM (third image). These two segmentation results then serve as an initialization for the level-set method with a developed penalty term Ψ (9) for a more accurate final segmentation (fourth image). The ground truth is given as the fifth image for comparison.

Valley detection and SVM methods are widely exploited in image processing. They are used as proper classifiers for the types of defects considered in the paper. Applying deep learning methods in this scope would require a significantly larger training set with ground truth data which is laborious to produce. Thus those two methods are used to classify the types of defects that they are best suited for, and a level set is used to aggregate their outcomes and produce a final segmentation map.

SVM is used to separate objects based only on color features. So its segmentation results are reliable only for paint, rust, and flaking objects. The paint crack can be “colored” in underlying metal color (Table 2, input image in the first row), or it can be “colored” in rust color (Table 2, input image in the fourth row), making color feature-based SVM unreliable as it segments crack region only when it has color features of rust damage. Therefore for fine crack segmentation, a valley detection method was used separately. The proposed level-set method combines the output of these two approaches to produce refined segmentation of all three types of damages.

Table 6 illustrates the influence of the kernel K shape on final segmentation results. In all the above experiments, the uniform disk-shaped averaging kernel was used. This simple type of kernel was chosen to simplify the understanding and interpretability of penalty term Ψ behavior. For practical application, the shape of the kernel K can be arbitrarily chosen depending on the segmentation objectives. In particular, Table 6 demonstrates the changes in the final segmentation map depending on the radius r of the kernel K.

Obtained results confirm the effectiveness of penalty term Ψ for the proper segmentation of defect type that is not distinguishable by color features only.

## 4. Discussion

Complex inspection of steel surfaces with protective coating requires the detection of several possible defect types (they can be metal or coating related, and also both simultaneously). Such inspection can be fulfilled automatically using image segmentation techniques. Thus in the case of multiple defect types, one should consider using multiclass image segmentation approaches. When dealing with the segmentation of multiple defects within one framework, it is possible to encounter the need to construct a more complex feature space than in the case of dealing with one single defect type. In this paper, we consider the segmentation of three defect types. The task’s difficulty is that using only color features causes some defects to overlap in feature space, thus leading to wrong segmentation. One way to overcome this problem is to increase the number of features that allow correct segmentation. In this paper, we proposed an approach that uses different methods of segmentation that are best suited for respective classes and combines their outputs for final refined segmentation.

As proposed in this work, multiclass level-set technique allowed us to build an efficient segmentation method for the simultaneous segmentation of three types of defects. The developed technique can be used to monitor infrastructure objects for complex rust damage detection and protective coating damage detection. The latter can be helpful as it allows the detection of areas with protective coating breakdown that can lead to further metal corrosion.

The developed technique combines two preliminary segmentation approaches best suited for detecting defects of certain types. SVM is used as an optimal classifier to distinguish between rust damage, paint flaking, and “healthy” regions of protective coating based solely on RGB pixel values of the input image. The valley detection approach was used solely for the paint crack detection. The segmentation output of those two algorithmically different classifiers was combined through the multiclass level-set approach with the specially designed penalty term Ψ. Such architecture made it possible to build an effective segmentation technique based on a simple feature set.

The final model contains a specific set of adjustable parameters and thus can be suited for different tasks. It also allows the use of different preliminary classifiers and penalty terms to be used for different image segmentation tasks.

Numerical experiments demonstrated good segmentation accuracy validated on ground truth data provided by the expert. It can be concluded from experimental results that the majority of segmentation errors are due to oversegmentation of defects. Mostly proposed approach suffers from oversegmentation of paint cracks and rust damages. In some cases, the proposed penalty term contributes to the overall error, as seen from the segmentation results in Table 2, rows 3 and 5. In segmentation results (row 3), in comparison to ground truth data, rust damage is completely absent; this is because the proposed term Ψ in the current setup tends to suppress rust damages under a certain size in favor of the present paint crack. A similar situation is in segmentation results in row 5, where the term Ψ “turned” part of the supposed crack defect into paint flaking. Those two errors are not to be considered major. The damaged area was segmented as such, only marked with a different type of defect. Unfortunately, the lack of research dedicated to the simultaneous segmentation of defects of multiple types makes it hard to conduct an extensive comparison against similar methods.

By choosing kernel shape K and functions $P_{kl}$ for different configurations of defect intersection, one can influence the final shape of the final segmentation maps and thus adjust the model to the desired behavior.

In general, the designed shape of term Ψ successfully overcame ambiguity introduced by preliminary classifiers and led to overall satisfactory results. The flexibility of the model (9) allows further development and improvement. It should also be noted that term Ψ is a label penalty term, thus, it is computed at each iteration which adds to the computational load of the method.

The advantage of the proposed technique is that it requires a considerably smaller training set for constructing efficient image segmentation methods in contrast to the heavily used nowadays deep learning approaches [97]. It is particularly useful in cases where the acquisition of large training data is complicated. Basically, neural networks, such as ResNet, UNet VGG, and LeNet need training data that include input data and corresponding output ground truth data. If it is required to modify the desired output, even the slight one, the training procedure should be repeated (or continued) with modified training data. Our approach and its mathematical foundation have several model parameters that can be adjusted manually in the desired direction (for example, Table 6) without the need to modify the training set and retrain the model. Of course, such parameter adjustment has its bounds and not always can replace retraining of the model with new data. Moreover proposed approach can incorporate the output of the neural networks in the manner it uses the output of SVM.

Nevertheless, with the proper training set, one can train suitable for the task CNN architecture to outperform the accuracy presented in this paper’s results. However, the proposed model is preferable in terms of flexibility and readjustment of the model for different types of defects and background.

The proposed method of image processing should also be used to diagnose chromium cathode coatings applied to a steel base in an electrolyte by the galvanic method since corrosion products are formed in their pores [98,99].

## 5. Conclusions

This article describes a developed multiclass level-set-based image segmentation approach for detecting multiple paint and steel-related defects. The developed approach uses the proposed label penalty term in level-set method to properly combine segmentation results of different types of preliminary classifiers (that also use different features) for a proper final segmentation map. Level-set with proposed penalty term can be used for different types of preliminary classifiers and object classes. Given the nature of segmented objects, the proposed approach showed high segmentation accuracy.

The disadvantage of the method is that the absence of data terms in model (9) contributes to the resulting error mainly through the over-segmentation of damages. The introduced penalty term Ψ in some spatial configurations of defects shifts the border between some defects in favor of one over another, thus, increasing the overall error.

Experimental results showed the approach’s viability for real-world application for automation of infrastructure object inspection for rust and protective coating damages.

Further development of the model can include incorporating data terms into the model (9). It supposedly should increase the accuracy of the final segmentation map. Also, further investigation can be applied to penalty term Ψ behavior for different parameter settings and adaptable kernel K shape and size.

It is also planned to apply the proposed method to study the corrosion destruction of chrome-plated steel.

## Figures and Tables

**Figure 1 sensors-22-07600-f001:**
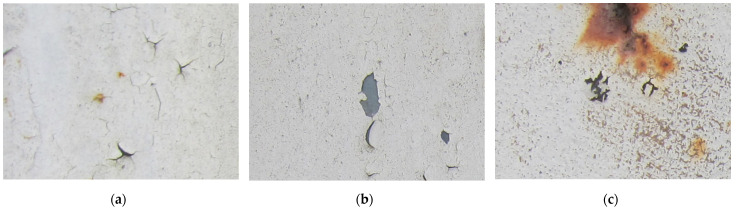
Examples of images of coated surface damages: (**a**) rust and paint cracking; (**b**) paint flaking and cracking; (**c**) rust damage and paint flaking.

**Figure 2 sensors-22-07600-f002:**
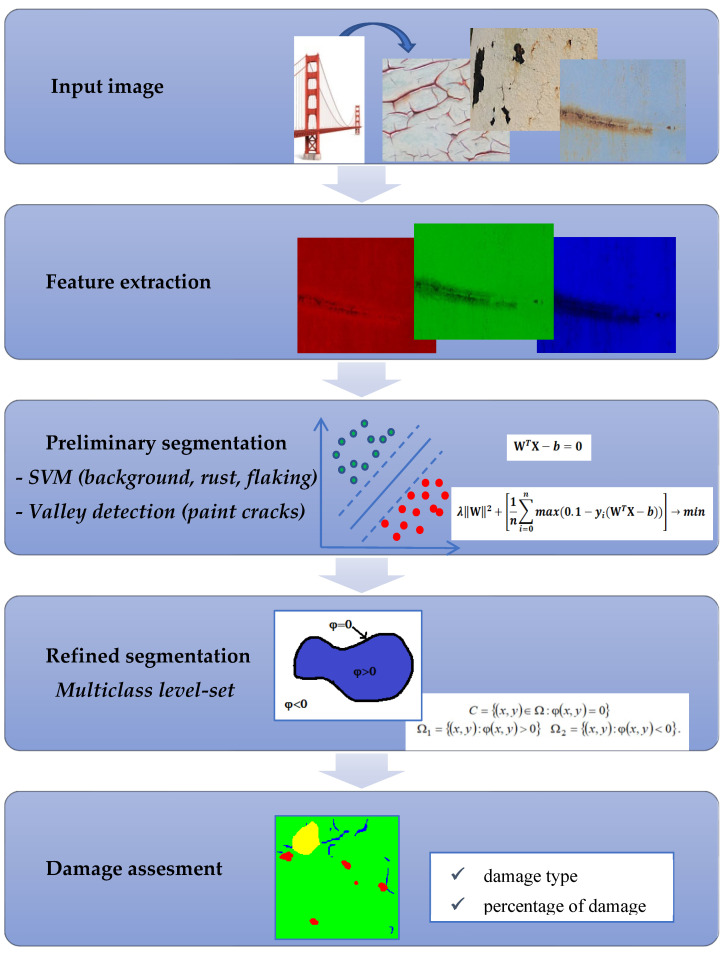
Flowchart of the proposed multiple type damage detection techniques.

**Figure 3 sensors-22-07600-f003:**
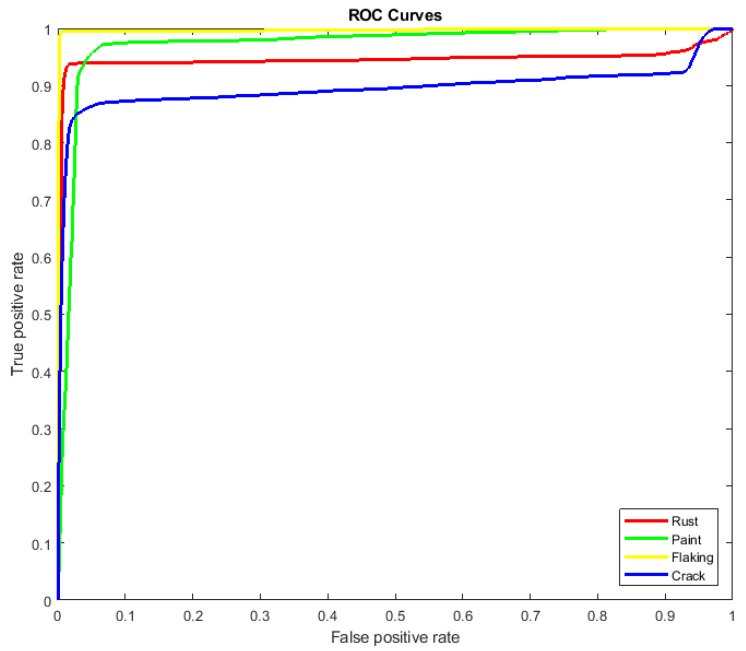
ROC curves generated for all classes: red for rust damage, green for paint, yellow for flaking, and blue for paint cracks.

**Table 1 sensors-22-07600-t001:** Error rates and segmentation accuracy of the developed approach.

Value	Rust	Background Paint	Flaking	Cracking	Overall
Error (%)	1.31	3.24	7.82	6.49	9.43
AUC (%)	94.49	97.19	99.70	89.54	–

**Table 2 sensors-22-07600-t002:** Examples of ground truth data of rust segmentation.

Type of Paint Coating Damage on Steel	Input Images	Ground Truth Segmentation	Segmentation by the Proposed Approach
Cracking of paint coating	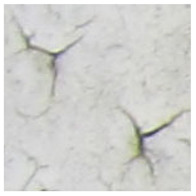	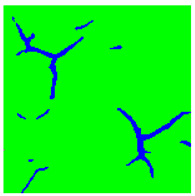	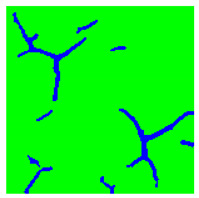
Rust	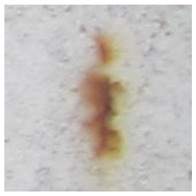	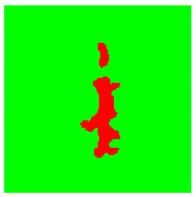	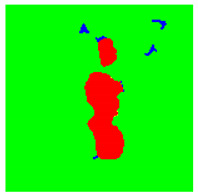
Flacking of paint coating, cracking, and rust	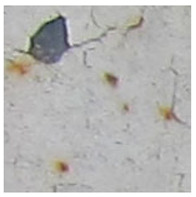	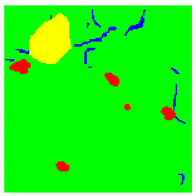	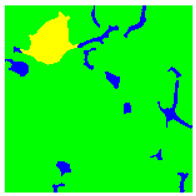
Cracking of paint coating	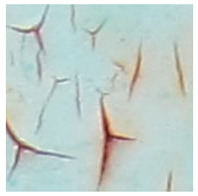	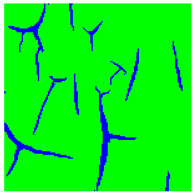	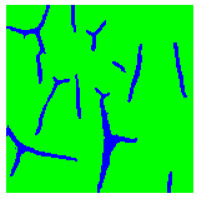
Cracking of paint coating and rust	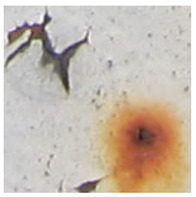	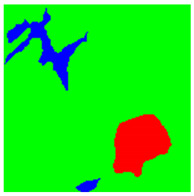	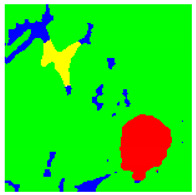

**Table 3 sensors-22-07600-t003:** Results of paint damage and rust segmentation.

Type of Paint Coating Damage on Steel	Input Images	Segmentation Results by the Proposed Model
Cracking of paint coating and rust	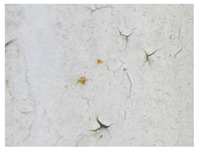	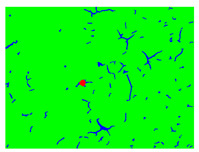
Rust damage	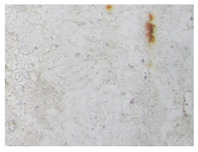	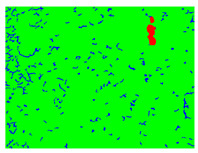
Cracking of paint coating and rust	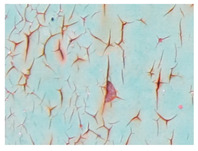	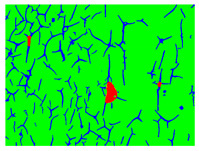
Cracking of paint coating and rust	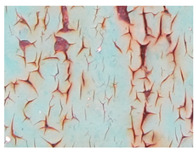	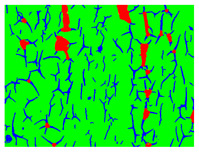
Flacking and cracking of paint coating	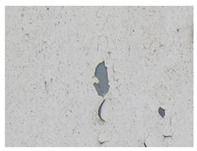	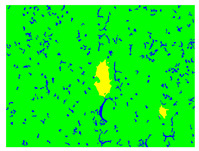

**Table 4 sensors-22-07600-t004:** Examples of valley detection for different threshold values *T*.

Input Image	Valley Filter Response	*T* = 128	*T* = 155	*T* = 165
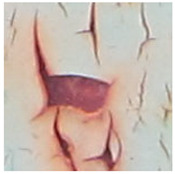	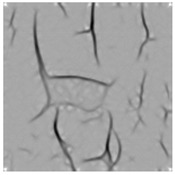	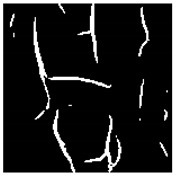	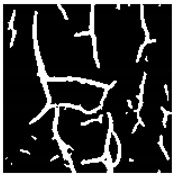	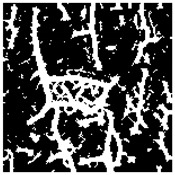

**Table 5 sensors-22-07600-t005:** Input image segmentation stages.

Input Image	Valley Detection Result	SVM Segmentation Result	Segmentation Result by the Proposed Method	Ground Truth
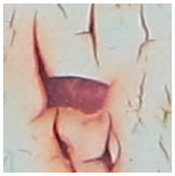	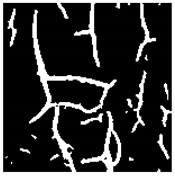	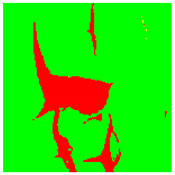	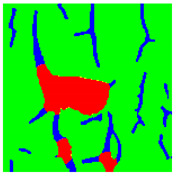	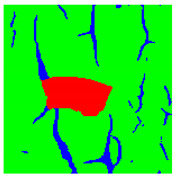

**Table 6 sensors-22-07600-t006:** Input image (Table 5) segmentation results for different radiuses of the kernel *K*.

*r* = 5	*r* = 11	*r* = 15	*r* = 21	*r* = 31
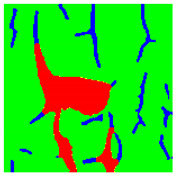	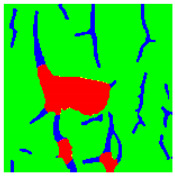	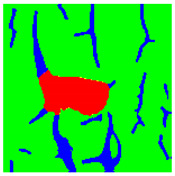	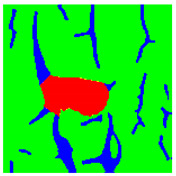	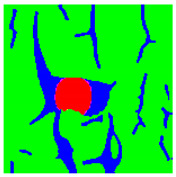

## Data Availability

The data presented in this study are available on request from the corresponding author.

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
