# Peer review of "Multiclass Level-Set Segmentation of Rust and Coating Damages in Images of Metal Structures"

_sensors, 2022, doi:10.3390/s22197600_

Round 1
Reviewer 1 Report
I think it’s interesting research. I would recommend it if the following issues are handled properly.
1. Deep learning has achieved great breakthroughs in imaging fields. Highlight the value of your algorithms of SVM and valley detection at this time.
2. Discuss the differences between your work and ResNet for image recognition.
3. In Table 5, the Segmentation result by the proposed method discerns the fine cracks compared with the SVM segmentation result. Please propose the possible reason.
4. Add more details about the feature extraction.
5. Add more information about your datasets.
6. Double check your formulas.
Author Response
We would like to thank the reviewer for their detailed comments and suggestions for the manuscript. We believe that the comments have pointed out important areas which required improvement. We hope that after completion of the suggested edits, the revised manuscript has benefitted from an improvement in the overall presentation and clarity. Below, you will find a point by point description of how each comment was addressed in the manuscript. Original reviewer comments have bold font, responses – regular font. Inserts in the text of the article are highlighted in green.
- Deep learning has achieved great breakthroughs in imaging fields. Highlight the value of your algorithms of SVM and valley detection at this time.
Thank you for the suggestion. We added paragraph with highlighted values of our algorithm after Table 5.
Valley detection and SVM methods are widely exploited in image processing. They are used as proper classifiers for the types of defects considered in the paper. The application of deep learning methods in this scope would require significantly larger training set with ground truth data which is laborious to produce. Thus those two methods are used to classify the types of defects that they are best suited for and level-set is used to aggregate their outcomes and produce final segmentation map.
- Discuss the differences between your work and ResNet for image recognition.
We added paragraph with differences between our work and ResNet as well as other neural networks.
Basically neural networks, such as ResNet, UNet VGG, LeNet need training data that include input data and corresponding output ground truth data. In case where it is required to modify the desired output, even the slight one, the training procedure should be repeated (or continued) with modified training data. Our approach and its mathematical foundation have a number of model parameters that can be adjusted manually in the desired direction (for example Table 6) without the need to modify training set and retrain the model. Of course such parameter adjustment has its bounds and not always can replace retraining of the model with new data. Moreover proposed approach can incorporate the output of the neural networks in the manner it uses output of SVM.
Nevertheless, with the proper training set one can train suitable for the task CNN architecture to outperform the accuracy of presented in this paper results. But in terms of flexibility and readjustment of the model for different types of defects and background the proposed model is preferable.
- In Table 5, the Segmentation result by the proposed method discerns the fine cracks compared with the SVM segmentation result. Please propose the possible reason.
Thank you for the suggestion. We added paragraph with explanation.
SVM is used to separate objects based only on colour features. So its segmentation results are reliable only for paint, rust and flaking objects. The paint crack can be “coloured” in underlying metal colour (Table 2, input image in the first row) or it can be “coloured” in rust colour (Table 2, input image in fourth row) making colour features based SVM unreliable as it segments crack region only when it has colour features of rust damage. Therefore for fine crack segmentation a valley detection method was used separately. The proposed level-set method combines the output of these two approaches to produce refined segmentation of all three types of damages.
- Add more details about the feature extraction.
We added lines with explanation.
For that purpose we used three-dimensional feature , where and are “red”, “green” and “blue” values of pixel colour components from RGB colour model respectively.
- Add more information about your datasets.
We added lines with additional information about dataset.
The proposed approach suggests combination of machine learning and image pro-cessing methods for multi-class segmentation of different defect types. Development, testing and validation of the considered approach were done based on the dataset of images of different defect types.
We used a set of 150 images with resolution of 480x640. The images included in the database show coatings of different colours, as well as different types of their damage - coating crack, coating flaking and rust damage. There are images with one type of defects presented on it as well as images with multiple types of damages in the dataset.
- Double check your formulas.
Thank you for valuable comment, we corrected formulae (1), (4), (7), (9), (10) and added explanations to formulae (1), (2), (4).

Reviewer 2 Report
This paper describes the combined detection of coating and rust damages on painted 18 metal structures through the multiclass image segmentation technique. Address the following comments
-. As 25 for the paint cracking, colour features are insufficient for separating it from other defects. For 26 efficient paint cracking detection we use the valley detection approach. This statement does not have clarity. Please refine it.
- many references just cited without explaining their work briefly like 15,16,17, 20, 21, 22 so on. Please explain briefly about each reference.
- Our previous works 70, 71, 72, 73, 74] were dedicated to rust damage segmentation under conditions that can distort its damage percentage assessment. Please explain them with atleast one statement and refer this analysis of dimentionality reduction techniques on big data
-what is the proposed method here. Its not explained clearly and no proposed architecture.
-explain all attributes used in the equations
-Complex automated inspection of steel surfaces with protective coating requires the detection of several possible defect types (they can be metal or coating related, and also both at the same time). How did you take automatic inspection on steel surfaces.
-no clarity on proposed method to understand so please explain it clearly and give the future direction
Author Response
We would like to thank the reviewer for their detailed comments and suggestions for the manuscript. We believe that the comments have pointed out important areas which required improvement. We hope that after completion of the suggested edits, the revised manuscript has benefitted from an improvement in the overall presentation and clarity. Below, you will find a point by point description of how each comment was addressed in the manuscript. Original reviewer comments have bold font, responses – regular font. Inserts in the text of the article are highlighted in green.
This paper describes the combined detection of coating and rust damages on painted 18 metal structures through the multiclass image segmentation technique. Address the following comments
-. As 25 for the paint cracking, colour features are insufficient for separating it from other defects. For 26 efficient paint cracking detection we use the valley detection approach. This statement does not have clarity. Please refine it.
We added lines with clarifications.
As for the paint crack, colour features are insufficient for separating it from other defect types as they overlap with the other three classes in RGB colour space. For preliminary paint crack segmentation we use the valley detection approach, which analyses the shape of defects. Multi-class level-set approach with a developed penalty term is used as a framework for refined final damage segmentation stage.
- many references just cited without explaining their work briefly like 15,16,17, 20, 21, 22 so on. Please explain briefly about each reference.
It was fixed: (inserts in the text of the article are highlighted in green).
Design, technological, and operational methods are used to provide long-term operation of coated products. Design methods include the selection of base and coating materials, conduction of corrosion [15, 16, 17] and tribocorrosion tests [18]. In particular, the scientists [15, 16] apply microelectrochemical studies of the stressed state of the metal on its structural components and crack-like defects, and in [17] the corrosion behavior of coated carbon steels in salt water is investigated.
Porosity studies of coatings [19] and tribological tests are carried out using computer image processing [20, 21, 22]. For example, researchers [20] used the image processing technique to develop a method of tribotechnical testing of coatings under conditions of wear by unfixed abrasive; and in works [21, 22], a photogrammetric approach was used to study the wear process and estimate volume loss. Strength tests [23, 24] as well as modeling and performance of thermal calculations of coatings [25, 26] are also performed.
- Our previous works 70, 71, 72, 73, 74] were dedicated to rust damage segmentation under conditions that can distort its damage percentage assessment. Please explain them with atleast one statement and refer this analysis of dimentionality reduction techniques on big data
Thank you, we added explanation to those references which were not explained.
Our previous works [70, 71, 72, 73, 74] were dedicated to rust damage segmentation under conditions that can distort its damage percentage assessment. They were focused on the detection of one type of damage – rust. In particular, the rust detection based on HSV image colour model [71], the use of the single-scaled retinex method [72], the influence of colour restoration based on a colour checker [73], and reduction of shadow effects [74] algebraic model with an asymmetric characteristic [75] and methods of logarithmic type image processing [76] as well as the application of inhomogeneity inforced piecewise smooth model [77] for image segmentation were considered. A dimensionality reduction PCA technique was applied in [78] for the rust segmentation in images obtained in irregular lighting conditions.
-what is the proposed method here. Its not explained clearly and no proposed architecture.
It was fixed. Corrected architecture is presented in Figure 2. We clarified description in the next paragraphs.
The main innovation of this paper is the introduction of the label-dependent penalty term that incorporates segmentation results of multi-class SVM and valley detection methods for refined resulting multi-class segmentation. To the best of our knowledge, it is the first time that three types of damages (coating crack, coating flaking and rust damage) are detected within one computational framework.
The first stage is a primal segmentation of input image. For this purpose two different approaches are used:
- a) The SVM is trained in a supervised learning mode based on a previously labelled training set with ground truth data for rust, flacking, and background classification. SVM was chosen for its ability to build an optimal decision hyperplane that separates classes. In this work, we use multiclass SVM with one against all approach. It is used as a primal segmentation method only for three-class classification (background, rust, and flaking). These three classes are discriminated by multiclass SVM because their features do not overlap in RGB colour space.
- b) We used a different approach for the primal segmentation of paint cracks. The reason for that is that they do not have unique colour features. As a result paint cracks in RGB colour space overlap with the other three classes making colour features insufficient for its classification. The characteristics that distinguish paint cracks are more of a geometric nature. We use a modified "valleys" detection method as a primal segmentations method for that purpose.
The second stage of approach suggests fusion of outputs of the first stage to produce final segmentation map.
-explain all attributes used in the equations
It was fixed. We added missing explanations of attributes to equations (1), (4).
-Complex automated inspection of steel surfaces with protective coating requires the detection of several possible defect types (they can be metal or coating related, and also both at the same time). How did you take automatic inspection on steel surfaces.
Thank you. The remark is valid. We meant that the segmentation parameters are selected without operator intervention not that all inspection is automatic. We corrected it by elimination the word “automated”.
-no clarity on proposed method to understand so please explain it clearly and give the future direction
It was fixed. We corrected architecture in Figure 2 and clarified description in the next paragraphs. See previous comment.
Future directions are given in the end of discussion and conclusions.

Round 2
Reviewer 2 Report
authors addressed all comments in the revision
Author Response
Dear Reviewer, Thank you very much for your time and reviewing our article.Authors